# Nanoparticle Enhancement of Natural Killer (NK) Cell-Based Immunotherapy

**DOI:** 10.3390/cancers14215438

**Published:** 2022-11-04

**Authors:** Dhanashree Murugan, Vasanth Murugesan, Balaji Panchapakesan, Loganathan Rangasamy

**Affiliations:** 1School of Biosciences & Technology (SBST), Vellore Institute of Technology (VIT), Vellore 632014, India; 2Drug Discovery Unit (DDU), Centre for Biomaterials, Cellular and Molecular Theranostics (CBCMT), Vellore Institute of Technology (VIT), Vellore 632014, India; 3School of Advanced Sciences (SAS), Vellore Institute of Technology (VIT), Vellore 632014, India; 4Small Systems Laboratory, Department of Mechanical Engineering, Worcester Polytechnic Institute, Worcester, MA 01609, USA

**Keywords:** nanoparticles, NK cells, NK cell-mediated immunotherapy, applications of nanoparticles, NK cell receptors

## Abstract

**Simple Summary:**

Natural killer cells are a part of the native immune response to cancer. NK cell-based immunotherapies are an emerging strategy to kill tumor cells. This paper reviews the role of NK cells, their mechanism of action for killing tumor cells, and the receptors which could serve as potential targets for signaling. In this review, the role of nanoparticles in NK cell activation and increased cytotoxicity of NK cells against cancer are highlighted.

**Abstract:**

Natural killer (NK) cells are one of the first lines of defense against infections and malignancies. NK cell-based immunotherapies are emerging as an alternative to T cell-based immunotherapies. Preclinical and clinical studies of NK cell-based immunotherapies have given promising results in the past few decades for hematologic malignancies. Despite these achievements, NK cell-based immunotherapies have limitations, such as limited performance/low therapeutic efficiency in solid tumors, the short lifespan of NK cells, limited specificity of adoptive transfer and genetic modification, NK cell rejection by the patient’s immune system, insignificant infiltration of NK cells into the tumor microenvironment (TME), and the expensive nature of the treatment. Nanotechnology could potentially assist with the activation, proliferation, near-real time imaging, and enhancement of NK cell cytotoxic activity by guiding their function, analyzing their performance in near-real time, and improving immunotherapeutic efficiency. This paper reviews the role of NK cells, their mechanism of action in killing tumor cells, and the receptors which could serve as potential targets for signaling. Specifically, we have reviewed five different areas of nanotechnology that could enhance immunotherapy efficiency: nanoparticle-assisted immunomodulation to enhance NK cell activity, nanoparticles enhancing homing of NK cells, nanoparticle delivery of RNAi to enhance NK cell activity, genetic modulation of NK cells based on nanoparticles, and nanoparticle activation of NKG2D, which is the master regulator of all NK cell responses.

## 1. Introduction

Our immune system is a living, breathing medicine that protects the body from various diseases, including cancer [1,2]. Three types of immunity have been identified: innate, adaptive, and passive immunity. Both innate and adaptive immunity systemically fight against the development of neoplastic cells and bacterial and viral infections in the body on a constant basis, which we term “immunosurveillance” [3].

The innate immune system can identify various pathogenic organisms through a molecular immunosurveillance process called pathogen-assisted molecular patterns or “PAMPs” that are not produced by the host (e.g., polysaccharides) [4,5,6]. PAMPs are evolutionarily conserved molecules that are present only in pathogenic organisms. They are recognized by the binding of PAMPs to cell surface and endosomal receptors (e.g., Toll-like receptors) [7]. Immune cells such as macrophages, dendritic cells, NK cells, and some epithelial cells, which form a network of immunosurveillance processes, recognize PAMPs [8]. The second mechanism by which the immune system is alerted during endogenous immunosurveillance is through danger-assisted molecular patterns or “DAMPs” (e.g., calreticulin or CD91) [4].

Cells that are physically, chemically, and biologically stressed or damaged/dying, which can be due to pathogens or other stressors (e.g., chemo or radiation), emit danger signals called DAMPs that alert the immune system [4]. DAMPs can be highly diverse, and their diversity is responsible for immune stimulation, immune modulation, and producing systemic antitumor immunity [4]. Together, PAMPs and alarmins form a network family of DAMPs that aids the immunosurveillance process [9]. Still, cancer cells evade the immunosurveillance process in several ways: loss of cell-adhesion antigens, impairment of cytotoxic T-lymphocytes and NK cells, generation of ligands that block recognition, and secretion of cytokines (e.g., VEGF, IL-10, and TGF-β) that inhibit maturation of dendritic cells [10,11]. In many cancers, tumor growth is supported through immunosuppression that hampers the effective antitumor response and tumor eradication. The tumor microenvironment is often “cold” or “desert-like” meaning they don’t secrete any molecules that could be identified by immune cells [12]. Additionally, tumor-associated macrophages (TAMs) and myeloid-derived suppressor cells (MSDC) envelop the tumor mass, which effectively keeps the infiltration of T-cells at bay [13]. TAMs also keep the immune system suppressed through the following additional pathways: regulation of PD-1 and CTLA-4 [14], T-cell suppression and exhaustion [15], recruitment of T_regs_ through CCL2 [16], and promotion of inflammation of the tumor microenvironment [16]. TAMs also secrete large amounts of cytokines, such as TGF-β and IL10^+^_,_ that can impair the cytotoxicity of NK cells [17]. Inflammatory cytokines induced by regulatory or suppressive immune cells promote cancer cell proliferation and suppress the antitumor immune response in the tumor microenvironment. Cells expressing indoleamine 2,3-dioxygenase (IDO) can inhibit the T cell response and lead to immunosuppression of the TME [18,19,20]. Arginase is produced by myeloid-derived suppressor cells and is released into circulation in patients with cancer [21,22]. These enzymes deplete the amino acids necessary for the proper functioning of T-cells. Together, the TME, along with immunosuppressive cells and inhibitory molecules, produces a significant barrier preventing immune attack, making it difficult to treat cancer. This explains some of the reasons for the failure of immunotherapy in many patients.

One of the mechanisms of tumor cell escape and subsequent metastasis involves immunoediting of the TME [3,23]. Cancer immunoediting is a dynamic process involving the interplay between tumor cells and immune cells where the tumor cells become less immunogenic over a period of time and develop the capability to escape [24]. Immunoediting can result in protection against cancer through immune-cell-mediated destruction of cancer cells, but it also can result in the escape of cancer cells beyond immune system control, which can lead to proliferation and metastasis. Three phases or “Es” of immunoediting have been identified: elimination, equilibrium, and escape [25] (Figure 1).

The elimination phase presents a phase where a strong immune response is observed due to a combination of both adaptive and innate immunity. This immunoediting phase is beneficial for systemic protection against cancer. Tumor cells have a high expression of surface calreticulin that acts as “eat me” signals for the dendritic cells [26,27,28,29]. Dendritic cells will process tumor antigens and present them on the surface to educate T and NKT cells, which is one of the most critical steps for the success of any form of immunotherapy [30,31]. The effector cells secrete IFN-γ; anti-angiogenic chemokines such as CXCL9, CXCL10, and CXCL11; and cytokines like IL-12 and IFN-γ that enhance macrophage polarization and NK cell activation [32]. Furthermore, Langerhans cells (LCs) secrete IL-15 and can cause the activation of NK and CD8+ T cells [33,34,35].

The second phase is the equilibrium phase. As the name suggests, it creates a microenvironment that contains a balance of pro-inflammatory cytokines and anti-inflammatory cytokines [36]. This phase of immunoediting results in dormancy of tumor growth. Nothing happens in this phase; the tumor is not eradicated, but it also does not grow. This phase is poorly understood, but it is known that the immune system keeps the tumor in a state of functional dormancy in this phase [36].

The third phase is called escape, and it occurs due to the following reasons: the loss of tumor antigens and subsequent loss of immune recognition of tumor cells [37,38]; an increase in secretion of cytokines such as TGF-β [39,40,41], IL-10 [42,43,44], and VEGF [45,46,47]; an increase in transcription factors such as STAT3 [48,49] and BACH2 [50,51]; and overexpression of molecules such as PDL1 and CTLA4, which act as a bridge to inhibit recognition by immune cells. Together, they provide an overarching barrier that prevents immune attack, and they impair the function of immune cells by inducing checkpoint blockade of the immune system. This environment provides enhanced opportunity for invasion and metastasis.

It is now known that cancers are immunogenic. The discovery of immune checkpoint blockade has revolutionized our understanding and created a new field of immuno-oncology. The initial discovery of PD-1 [52] and CTLA-4 [53] led to several promising therapeutic drugs (e.g., Ipilimumab, the first immunotherapy drug that targets CTLA4) for cancer treatment [54]. Ipilimumab blocks the CTLA-4 pathway, which is a T-cell inhibitor that results in the infiltration of cytotoxic T-cells into the TME [53]. Early investigations with Ipilimumab as a monotherapy in advanced metastatic melanoma were highly promising, showing an objective response rate of 40–60% for two years despite adverse side effects in 10–15% of patients [54,55]. The hefty price tag makes it hard to prescribe this as the first line of treatment for most individuals [55]. However, immunotherapy does not work for everyone, and only a fraction of patients benefits from immune-checkpoint-blockade-based therapies. The percentage of patients estimated to respond to checkpoint inhibitor drugs was 0.14% in 2011 and increased to 12.46% in 2018 [56]. Nanotechnology can assist with increasing the efficacy of immunotherapies in a variety of ways. The small size of nanoparticles, 10–100 nm, makes them ideal for modulating immunotherapies in a variety of ways that may not be possible using traditional antibody delivery approaches. These include delivery of immune checkpoint inhibitors (ICI) or antibodies to the TME using encapsulated nanoparticles, creation of tumor antigens through nanoparticle-mediated local ablation of cancer cells and necroptotic cell death, creation of designer nanoparticles with PAMPs and DAMPs that are customized to the TME, education of T cells and NK cells in vitro using nanoparticles based on the mutational status of the TME, delivery of pro-inflammatory cytokines using nanoparticles to neutralize the immunosuppressive environment, and many others. Thus, nanotechnology could potentially enhance immunotherapies and overcome the immunosuppressive environment in a variety of solid cancers. In this review, we focus on the roles NK cells and how nanotechnology can be used to modulate NK cell activity in tumor eradication.

### 1.1. Natural Killer Cells

Natural Killer (NK) cells are one of the first lines of defense against infections and neoplasms. NK cells are lymphocytes like B cells and T cells, that originate from common lymphoid progenitor cells. NK cells originate from bone marrow and from some secondary lymphoid tissues like the spleen, lymph nodes, or tonsils [57]. NK cells constitute 10–15% of circulating lymphocytes [58]. They are also regulatory cells and interact and communicate with dendritic cells, macrophages, T cells, and endothelial cells [59].

NK cells express MHC class I receptors, which are needed for self-tolerance and functional competence [60]. When encountering MHC class I-deficient hematopoietic cells (e.g., tumor cells), they can recognize the “missing self”. An immunological synapse is formed between the NK and target cell. The polarized lysosomes of the NK cell are activated, and they deliver their cytotoxic contents (e.g., granzymes and perforins) at the synapse, killing the target cell (Figure 2). Downregulation of HLA class I antigen expression in tumor cells also activates NK cells [61]. Many activating receptors have been identified in NK cells in the past decade, including NKG2D, SLAM, DNAX, and NCRs [62]. NK cells use a variety of receptors and pathways to sense their environment, synergistically communicate, and modulate their natural cytotoxicity. Not just one receptor activates NK cells. Only when multiple activating receptors are engaged, and information is acted on synergistically does the cytotoxic activity of NK cells come into effect [62]. This suggests that NK cells have a much more sophisticated control mechanism than T and B cells.

While early ICI-based immunotherapies have focused on T cell proliferation to eradicate the TME, and while recent FDA approval of CAR-T cells is promising, studies have revealed the following disadvantages of CAR-T cells: T cell exhaustion [63,64]; CAR-T cell-associated toxicities may be severe as normal cells surrounding the tumor may also express similar antigens [65]; the response is limited in solid tumors such as breast cancer (ORR: 19%) [66]; and that MHC class I-deficient cells can escape T cells, resulting in poor tumor infiltration by cytotoxic T cells [67]. Due to these shortcomings of T cells, NK cells have recently become more actively researched, with the aim of understanding their functions and controlling their response to selectively work against tumors.

In addition to the direct cytotoxic activity of NK cells, they could also convert the immunologically “cold” microenvironment into a “hot” microenvironment that could render immunotherapy effective.

**Figure 2 cancers-14-05438-f002:**
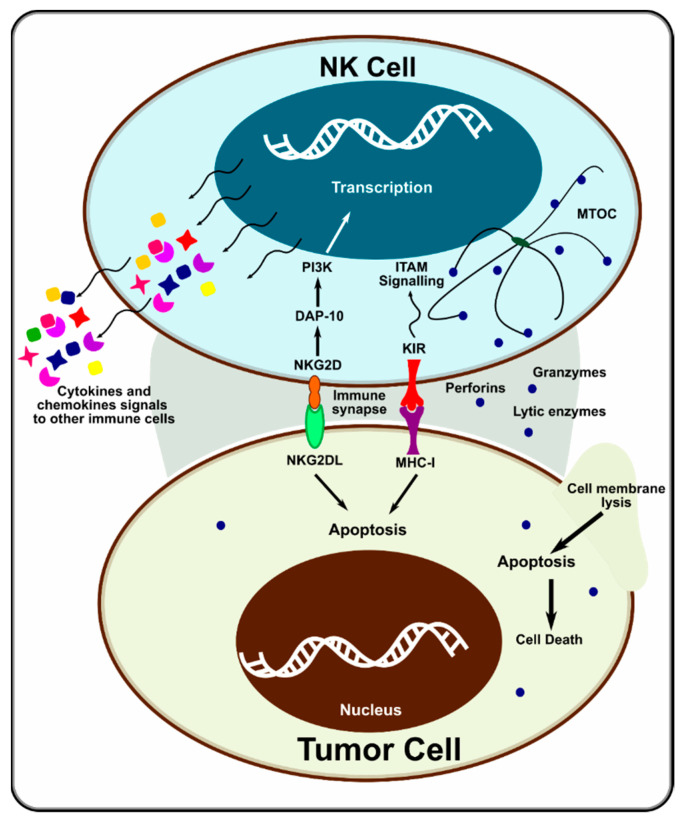
Mechanism of action of NK cell on tumor cells. Adapted from [68].

### 1.2. Potential NK Cell Activation Targets

NK cells belong to the innate immune system and thus do not require any prerequisite stimulation like T or B cells [69]. In NK cell-based immunotherapies, strategies should be developed to increase NK cell proliferation, expression levels of cytotoxic agents, infiltration capability, inhibition of inhibitory signaling pathways, and inhibition of NK cell exhaustion. Thus, there are two strategies to develop NK cell-based immunotherapies: to upregulate or activate molecules that would enable the activation of NK cells (such as activation of the NKG2D receptor or delivery of NK cell-activating cytokines (e.g., IL-2, IL-15, or IL-18)); and to inhibit molecules that restrict NK cell activity. For instance, NKG2D, DNAM1, NCR, CD16, NCAM/CD56, CD96, CXCR1, CX3CR1, IL-2/IL-15, CD11, and CCR7 are NK cell-activating receptors; while CD94/NKG2A, TIM-3, TIGIT, KIRs, LAG-3, and PD-1 are NK cell-inhibitory receptors [70]. A list of activating and inhibitory NK cell receptors and their functions is provided in a concise form in the table below (Table 1).

## 2. Application of Nanoparticles in NK Cell-Based Immunotherapy

The small size of nanoparticles compared to a cell is conducive for delivering drug molecules or cargo. Nanoparticles are not restricted to a delivery vehicle, and they play a pivotal role in the activation of different immune cells, including NK cells. This section of the review summarizes the potential of nanoparticles to assist NK cell-based immunotherapy. Based on the mechanism of action or the target of the immunotherapy, NK cell-based immunotherapies are further subdivided into the following five categories: nanoparticle-assisted immunomodulation to enhance NK cell activity, nanoparticle enhancing homing of NK cells, nanoparticles delivering RNAi to enhance NK cell function, nanoparticles for genetic modification of NK cells, nanoparticles activating NKG2D receptor (Figure 3). NK cell-based nano-immunotherapies are a nascent and developing field, and these therapies might move toward the clinical phase in the coming decades. Currently, CytoSen Therapeutics and KBI Biopharma have developed nanoparticle-based NK cell therapy, and it has entered phase two of a clinical trial [87]. Phase one clinical trials are ongoing for NK cell-based treatment of esophageal cancer [88]. PRECIOUS-01 is a natural killer T cell (iNKT) activating agent that consists of threitolceramide-6 (ThrCer6, IMM60) and New York Esophageal Squamous Cell Carcinoma-1 (NY-ESO-1) cancer-testis antigen peptides encapsulated in a biodegradable polymer poly (lactic-co-glycolic acid) (PLGA) nanoparticle [88]. NY-ESO-1 is overexpressed in many advanced cancers, including lung cancers, bladder cancers, melanoma, and ovarian cancers.

### 2.1. Nanoparticle-Assisted Immunomodulation for Enhanced NK Cell Activity

Immunomodulation-based cancer therapeutics have recently emerged, as this approach employs the host’s own defense mechanisms to recognize and eliminate cancerous cells. Nanoparticles acting as immunomodulators can improve therapeutic effects and overcome limitations of conventional methods of cancer treatment [92]. There is a broad scope of nanoparticles for drug delivery and an increase in immunotherapeutic efficiency. Some nanoparticles can be used to deliver anticancer drugs, chemokines, and cytokines [93,94,95]. Lipid-based nanoparticles could encapsulate these molecules and effectively deliver them to the tumor site [96,97,98]. Another strategy involves the use of surface-engineered nanoparticles; polymeric or metal-based nanoparticles can be modified such that tumor antigens or antibodies can be docked onto the surface (Table 2) [93,99,100]. For instance, liposome-loaded immunomodulatory agents like TGF-β and IL-2 has also been shown to enhance immune cell infiltration in the tumor microenvironment [101,102]. Similarly, the use of selenium-based nanoparticles that could enhance the function of NK cells has been reported [103]. Selenium-based nanoparticles not only enhance NK cell function but also induce non-specific humoral and cell-mediated immune responses and elevate IL-17 and IFN-γ [103,104,105]. Ruthenium nanoparticles were decorated with bispecific antibodies (SS-Fc, anti-CD16, and anti-CEA), and these triggered immune responses by stimulating NK cells to induce necrosis and apoptosis [106,107].

### 2.2. Nanoparticles Enhancing Homing of NK Cells

Homing is a mode of signaling between homing receptors present on immune cells and homing receptor ligands on the affected tissues or tumor-secreted molecules [128]. These interactions in a tumor microenvironment could recruit more effector immune cells and enhance immune cell infiltration [128,129]. Cancer cells secrete cytokines and chemokines like TNF-α and ROS, which attracts other effector immune cells. Depending on the function of the cytokines and chemokines, different immunological signals are triggered [130]. These signals could either induce inflammation or activate cytotoxic activity. Chronic inflammation induces tumor cell proliferation, survival, and metastasis [131]. Immune cells involved in inflammation include neutrophils, macrophages, and myeloid-derived suppressor cells [132]. Some of these cytokines induce effector immune cells that could kill cancer cells by activating T cells, B cells, and NK cells [133]. NK cells have a crucial role in the cytotoxicity-mediated killing of tumor cells, and they secrete various cytokines and chemokines that could recruit other immune cells [132]. Once NK cells infiltrate the tumor microenvironment, multiple signaling pathways are triggered by a ligand-receptor interaction, which results in the release of perforin, granzymes, and apoptosis-inducing factors [134]. Hence, it is essential to induce NK cell homing in the tumor microenvironment for effective tumor reduction.

The use of nanoparticles, especially magnetic nanoparticles, for homing immune cells has been explored extensively. For instance, the conjugation of iron oxide nanoparticles on the surface of primary NK cells would significantly enhance the homing of NK cells in the tumor microenvironment. These nanoparticles showed significant antitumor efficacy by increasing the expression of granzymes and perforins in neuroblastoma cells compared with unmodified NK cells [135]. Moreover, the killing of cancer cells by NK cells was associated with the homing of NK cells as well as by external magnetic guidance [135]. These nanoparticles have attracted significant interest for the delivery of immunotherapeutic drugs. Similarly, immunomodulatory magnetic microspheres also increased NK cell infiltration at the tumor site [136]. These microspheres consist of iron oxide nanocubes and recombinant interferon gamma encapsulated in biodegradable poly(lactide-co-glycolide). This architecture allows sustained release of IFN-γ and sensitive MRI T2 contrast agents, which enables both the homing of immune cells and MRI imaging. These microspheres were evaluated on an orthotropic liver tumor VX2 rabbit model, which showed enhanced proliferation and infiltration of NK cells. There are various strategies being adopted for the development of nanoparticle-based NK cell homing. Some of these strategies involve delivering effector molecules such as granzymes, membrane-bound heat shock proteins, and cytokines such as IL-12 using magnetic nanoparticles, which not only enhance homing but also activate NK cells or kill tumor cells (Table 3) [136].

### 2.3. Nanoparticles Delivering RNAi for Enhancing NK Cells Activity

RNA effectors such as siRNA, miRNA, and shRNA could silence specific genes, which could alter genomic function and enhance antitumor activity. The use of these RNA effectors to enhance NK cell activity is classified under RNAi-mediated immunotherapy [142]. A manganese dioxide (MnO_2_) nanoparticle system was used to deliver small interfering RNA (siRNA) targeting transforming growth factor-β receptor-2 (TGFBR2), which is known to inhibit the function of NK cells. These nanoparticles loaded with TGFBR2 siRNA protected NK cells from immunosuppression by inhibiting TGFBR2. Thus, silencing TGFBR2 in NK cells made the tumor microenvironment more immunoresponsive by activating NK cells. This suggests that these nanoparticles can be used to enhance the antitumor effects of NK cells through TGFBR2 knockdown and increased expression of IFN-γ [143]. Thus, RNAi-based adoptive NK cell therapy has great potential to improve the survival of cancer patients (Table 4).

EpCAM (epithelial cell adhesion molecule)-targeted cationic liposomes containing si-CD47 and si-PD-L1 were used to knock down immunosuppressive CD47 and PD-L1 [144]. These liposomes effectively prevented the growth of tumors and reduced lung metastasis in 4T1 tumor-bearing mice. These siRNAs could slightly increase the percentage of NK cells, promote the NK cell response, and also increase antibody production. It was also reported that this dual blockade of innate and adaptive immune checkpoint increased the expression of IFN-γ and IL-6 in vivo and in vitro. This suggests that this dual blocking system can be employed to stimulate both adaptive and innate immunity to fight breast cancer [144]. Similarly, another study utilized cationic lipid-assisted nanoparticles encapsulated with siCD155. These nanoparticles were efficient in delivering siCD155 to B16-F10 melanoma cells and macrophages in vitro and in vivo [145]. Downregulation of CD155 promoted the activation of NK and T cells while inhibiting the proliferation of melanoma cells. This suggests that nanoparticle-delivered siCD155 can be used to inhibit melanoma cell proliferation and reprogram the tumor microenvironment with proliferated NK cells and T cells [145].

A novel cocktail strategy was developed by combining NK cell-derived exosomes with miRNA-loaded biomimetic nanoparticles for targeting and therapeutic delivery of miRNA to neuroblastoma cells. NK cell-derived exosomes induced miRNA-loaded nanoparticles to leave systemic circulation and concentrate in tumor cells. One of the major advantages of using NK cell-derived exosomes is that these exosomes might have tumor-specific accumulation and may not be cytotoxic to normal tissues. It was demonstrated that when mice bearing CHLA-255-luc-induced tumors were treated with this cocktail, tumor growth inhibition was observed, which involved synergistic activity between exosomes and miRNA-loaded nanoparticles [95]. Another NK cell-derived exosome was utilized by Neviani et al., where they conjugated NK cell-derived exosomes with miRNA-186. These exosomes exhibited significant cytotoxicity against *MYCN* gene-amplified neuroblastoma cell lines. Moreover, the cytotoxicity was dependent on the expression of miRNA-186. In vitro studies revealed that these exosomes induced downregulation of TGF-β, which is involved in immune escape. These results suggested that NK cell-derived exosomes loaded with miRNA-186 are a promising therapeutic to promote cytotoxicity of NK cells and block immune escape by tumor cells [146].

### 2.4. Nanoparticles for Genetic Modification of NK Cells

Due to the dynamic role of NK cells in tumor identification and surveillance, adoptive immunotherapy is being developed as a next-generation therapeutic tool. Various strategies have been employed to improve the efficiency and number of NK cells in the tumor microenvironment (Table 5). Understanding NK cell biology and its interactions with the tumor microenvironment enable the modification of NK cells for better and relevant NK cell immunotherapy [147]. Nanoparticle-based delivery of chimeric antigen receptor genes to patient-derived NK cells could lead to the development of CAR-NK cell-based therapy (Figure 4).

Chitosan nanoparticles loaded with IL-2 and NKG2D genes activated NK cells and cytotoxic T cells in vitro [71]. They also showed enhanced tumor accumulation due to the enhanced permeability and retention (EPR) effect and the gathering of lymphocytes in the tumor microenvironment [71]. Consistent with the in vitro results, reduced tumor volumes and improved survival time was observed in CT-26 tumor-bearing mice. These results suggested that nanoparticle-mediated delivery of IL-21 stimulated the antitumor effects of NK cells efficiently, which led to enhanced antitumor activity [71]. In another independent study, chitosan nanoparticles successfully delivered IL-15 and NKG2D genes into cancer cells [148]. These nanoparticles, containing genes for the NKG2D-IL-15 fusion protein, bind to the NKG2D receptor of cytotoxic T cells and NK cells and activate them. IL-15 and *NKG2D* fusion proteins enhanced the antitumor immune response in B16BL6 melanoma cells by activating >5% of cytotoxic T cells and >50% of NK cells. Moreover, they also showed reduced tumor volume and enhanced survival time in B16BL6 tumor-bearing mice. This suggested that these nanoparticles can be used as a fusion gene vaccine for immunomodulation and tumor growth suppression [148].

An effective method to destroy tumor cells is to use engineered NK cells that consist of dendrimer-entrapped gold nanoparticles containing *human ferritin heavy chain (hFTH1)* gene-transfected NK cells [149]. These PEG-modified dendrimer-entrapped gold nanoparticles efficiently provided high-quality imaging of transfected NK cells. This system has *hFTH1* transfected effectively at a ratio of 5:1 to allow magnetic resonance imaging (MRI) of NK-92 cells and breast cancer cells. Furthermore, these nanoparticles guided NK cells toward the tumor environment for efficient gene therapy in 4T1 tumor-bearing mice. It was suggested that this system could be an efficient vector for genes and also for real-time monitoring with MRI [149]. Similarly, a multi-kinase inhibitor TUS2 (Tumor suppressor candidate 2) gene was delivered using nanovesicles [150]. TUS2 contributes to significant tumor growth reduction in a *Kras*-mutant syngeneic mouse lung cancer model. Furthermore, it was experimentally found that it increases the levels of circulating and splenic NK cells and CD8^+^ T cells, decreases the action of Treg cells and MDSCs, and reduces some of the checkpoint receptors such as PD-1, CTLA-4, and TIM-3 [150]. Later in the same study, the anti-PD-1 antibody and the TUS2 plasmid were loaded in the nanovesicle, which then revealed synergistic action against tumor cells via enhanced cytokine-based NK cell activation. A multifunctional magnetic nanoparticle system was synthesized, and it has been shown to be capable of enhancing NK cell function and tracking nanoparticles using magnetic resonance and fluorescence imaging. These multifunctional nanoparticles were designed by applying cationic polydopamine (PDA) coating and plasmid DNA to the surface of magnetic nanoparticles. The magnetic core enabled MRI of NK cells, and the cationic layer enabled them to serve as plasmid DNA carriers. This system showed better cytocompatibility and induced the expression of EGFR targeting chimeric antigen receptors. In vivo results showed that tumor volumes were reduced when treated with these multifunctional nanoparticles in the MDA-MB-231 xenograft mice model. This shows the excellent ability of these cytocompatible multifunctional nanoparticles to potentiate NK cell-mediated antitumor activity and to enable in vivo monitoring [151].

**Table 5 cancers-14-05438-t005:** Comprehensive information on nanoparticle-based genetic modification on NK cells.

Sr No.	Nanoparticles	Target NK Cell	Ligand	Mechanism of Activation	Effect on Tumor Cells	Ref.
1.	Chitosan nanoparticles comprised of extracellular *NKG2D* gene domains and *IL-21* gene	-	-	Increased expression of NKG2D receptor and IL-21	Reduced tumor volumes and improved survival time was observed in CT-26 tumor-bearing mice	[71]
2.	Chitosan nanoparticles comprised of extracellular *NKG2D* gene domains and *IL-15* gene	-	-	Increased antitumor immune response in B16BL6 melanoma cells by activating >5% of cytotoxic T cells and >50% of NK cells	Reduced tumor volume and enhanced survival time in B16BL6 tumor-bearing mice	[148]
3.	Dendrimer-entrapped gold nanoparticles containing *human ferritin heavy chain (hFTH1)* gene	-	-	hFTH1 guided NK cells to infiltrate near the tumor	Microspheres assisted in vivo MRI imaging to locate breast cancer pre- and post-treatment	[149]
4.	Magnetic nanoparticle coated with polydopamine containing plasmid DNA for targeting EGFR chimeric antigen receptor	EGFR	EGFR targeting CAR-NKs	Induced NK cell-mediated antitumor activity	Tumor volume reduction in MDA-MB-231 xenograft mice model	[151]

### 2.5. Nanoparticles Activating NKG2D Receptor

NKG2D is a C-type lectin-like activating receptor expressed on NK cells, NKT cells, and cytotoxic T cells. The ligands of the NKG2D receptor-retinoic acid early induced transcript-1 (RAE-1), H60, UL-16 binding protein like transcript-1, MHC-I chain-related protein A, and ULBPs. These ligands could initiate the NKG2D signaling pathway, which results in the activation, proliferation, and expansion of immune cells [71]. Many researchers have explored these ligands and other novel methods to activate NK cells.

NKG2D receptors initiate ITAM (tyrosine-based activation motif) signaling. Once the signal is triggered, co-stimulatory molecules CD28 and ICOS are triggered and lead to a cascade of reactions that culminate in the activation of PI3K, including reactions involving transcription factors [152]. Activation of these pathways requires targeted delivery of NKG2D ligands or certain cytokines, and nanoparticles mediate the delivery of these ligands. For instance, nanoemulsion of a TGF-β inhibitor and selenocysteine increases the lytic capability of NK cells by sensitizing NKG2D ligands [153]. The TGF-β inhibitor effectively restricted the TGF-β/TGF-β RI/Smad2/3 signaling pathway, which increased the concentration of NKG2D ligands on the tumor cell surface. Selenocysteine supports the expression of NKG2D receptors and suppresses PD-1 expression in γδ T cells [153]. Thus, these combinations enhance the activity of NK cells and act as an appropriate example of adoptive immunotherapy. Some of the other strategies involved are the use of zinc-doped vascular endothelial growth factor (VEGF) receptor-targeted super magnetic nanoparticles, which not only activate NK cells but also trigger magnetic hyperthermia [154]. Further methods for the activation of NK cells are mentioned in Table 6.

## 3. Limitations of Nanotechnology-Based NK Cell Therapy

As in any immunotherapy, nanotechnology-based NK cell therapy has several limitations. It has been reported that metallic nanoparticles (e.g., Cu) could impose hepatoxicity [157]. The disintegration of chemotherapeutic drugs is mostly dependent on the kidney, whereas degradation of nanoparticles is mostly dependent on the liver. In the process of liver detoxification, many of these nanoparticles accumulate in nearby tissues. This accumulation interrupts various other enzymatic degradation processes [158]. For this reason, it is highly recommended that during nanoparticle evaluation, potential hepatotoxicity is carefully assessed [159]. Similarly, the accumulation of Fe from nanoparticles could potentially cause oxidative stress and ferroptosis, which can potentially affect immune cells and therapy [160].

Although NK cell-based immunotherapies are promising, there are some limitations or challenges in utilizing this therapy. One of the most crucial issues is the uninterrupted supply of NK cells for treatment, which could make treatment very expensive. Clinical trials utilizing NK cells require large numbers, typically 5 × 10^6^ to 5 × 10^7^ cells per kilogram [161]. Sometimes it is difficult to obtain autologous NK cells, which may be due to the health condition of the patients [162]. NK92 cells are generally used in clinical trials and even for the development of CAR-NK cells. NK92 cells are transformed cell lines and thus could be easily enriched in laboratory conditions. However, there are changes in the tumorigenic conversion of these cells, which is a threat during the treatment regimen [163]. However, the use of induced pluripotent stem cells (iPSCs) in the generation of NK cells might be a future strategy to resolve these issues and to obtain large-scale production of NK cells for therapies [164,165]. Another problem is the identification of an appropriate tumor antigen for developing CAR-NK cells that target resistant tumor cells [162].

## 4. Future Prospects

Our immune system protects the body from a host of diseases, including cancer. The advent of immunotherapy offers the promise of cancer prevention and cure based on activation and control of the immune system. So far, however, the clinical results of immunotherapies have been mixed and work only for a fraction of people as monotherapies. Therefore, a combination approach is being attempted, which has seen incremental therapeutic success. It is becoming more apparent that there are potentially many reasons for immunotherapy failure, including T cell exhaustion, defects in homing molecules on the cell surface, increased MDSC cells in circulation, CAR-T damaging normal tissues, and the release of immunosuppressive cytokines and enzymes that impair T cells, NK cells, and other immune cells.

The idea of using nanoparticles is to force the TME to interact artificially (e.g., blocking CTLA-4) if conventional antibodies fail to elicit a response. Furthermore, nanoparticle-based drugs can also act as multiple lines of defenses by design. Nanoparticles, due to their size, can be designed into multiple different types: nanoparticles to release antigens by necroptotic killing of tumor cells to drive dendritic cell activation, better homing to the TME compared with immune cells, anti-inflammatory nanoparticles to minimize inflammation in the tumor environment in the absence of an immune response, and nanoparticles to aid the proliferation of NK cells. Many of these approaches must be combined for the complete eradication of the TME.

For example, necroptotic cancer cell mimetic nanovaccines have been demonstrated to contain those artificial tumor antigens to induce NKG2D+ NK cells and IFN-γ-expressing CD8+ T cells [166]. Similarly, synthetic vaccine nanoparticles using poly (γ-glutamic acid) tumor antigens and TLR-3 agonists have been demonstrated [167]. These vaccines were able to successfully generate the expansion and activation of NK cells and cytotoxic T cells [167]. It has been observed that certain cytokines such as IL-18, IL-12, IL-15, CD16, CD25, and NKG2A receptors can induce NK cell memories [168]. Nanoparticles inducing these receptors and cytokines could induce cytotoxicity, a higher tumor response, and could trigger memory-like NK cells [168]. Similarly, haptens can also induce memory-like NK cells, and hapten-decorated nanoparticles could be designed for NK cell-based immunotherapies [169]. Conversion of a cold tumor environment to a hot tumor environment is also one of the strategies employed in nano-immunotherapy. Cold tumor environment conversion sometimes includes increased infiltration of immune cells and increased in expression of MHC class I and II [170]. Thus, the synergistic role of nanoparticles could include the possibility of future on-immunotherapies.

## 5. Conclusions

In conclusion, the disadvantages of T cell-based therapies (namely T cell exhaustion resulting in an inadequate or failed therapeutic effect, CAR-T cell-associated toxicities, the limited response of T cell-based therapies in solid tumors such as breast cancer, and MHC class I-deficient cells escaping and resulting in metastasis) are the reasons for NK cell-based therapy development. NK cells are more sophisticated than T cells, as they can recognize “self” and “missing self”. Recognition of the “missing self” triggers a response in which NK cells kill the target cell using a synapse and deliver cytotoxic cargo. Conventional limitations of NK cells are the reduced lifespan of NK cells in the blood, low activity of NK cells in the suppressive TME, inadequate homing of NK cells, and limited interaction frequency of NK cells with tumor cells [171].

Nanoparticles have proved to possess great potential in the various preclinical studies that have assessed their ability to assist NK cell effectiveness. Nanoparticles could assist NK cells in many ways, including their activation, proliferation, the release of tumor antigens, and removing the chronic inflammatory environment of the tumor. Nanoparticles that assist immunotherapy are attractive due to their small size and diverse choice of nanoparticles. One could potentially influence several receptors on NK cells through nanoparticle–antibody homing, resulting in cytolytic activity. While much is to be understood about the interactions of nanoparticles with NK cells, the present research shows much promise in developing nanoparticle-based immunomodulatory drugs to further increase the effectiveness of immunotherapy.

## Figures and Tables

**Figure 1 cancers-14-05438-f001:**
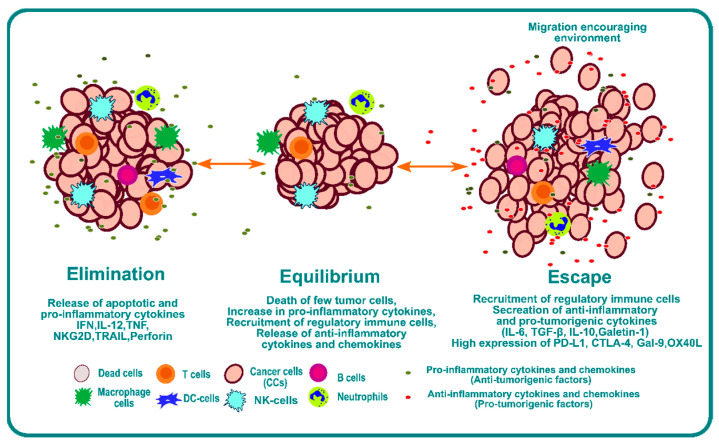
Schematic representation of the phases of immunoediting, namely elimination, equilibrium, and escape.

**Figure 3 cancers-14-05438-f003:**
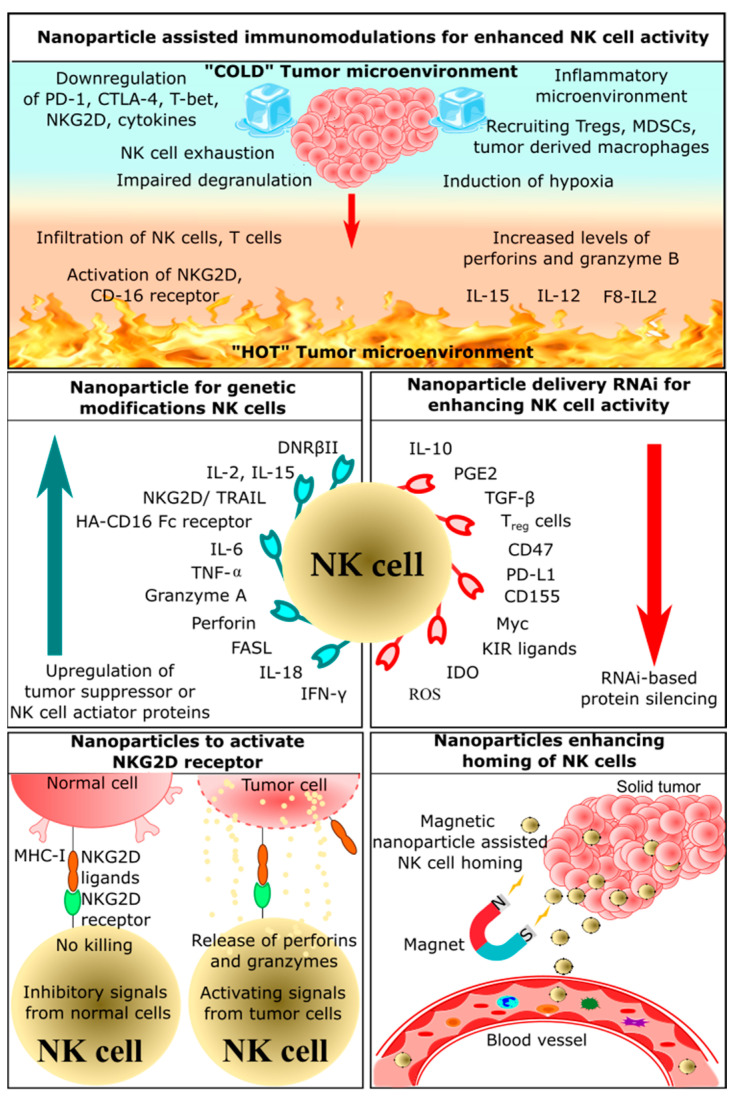
Schematic representation of various strategies for NK cell-based nano-immunotherapies. Adapted from [89,90,91].

**Figure 4 cancers-14-05438-f004:**
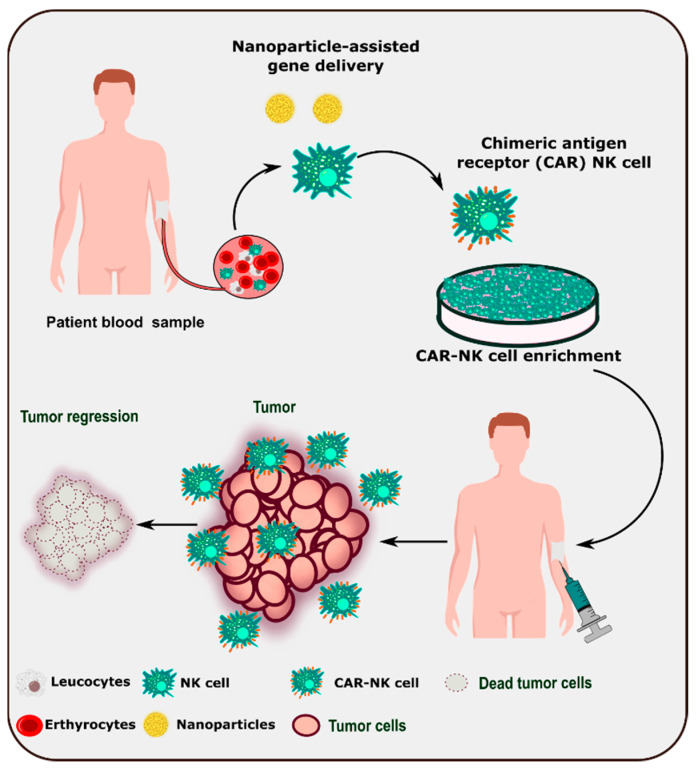
Schematic representation of nanoparticle-assisted development of CAR-NK cell-based therapy.

**Table 1 cancers-14-05438-t001:** Comprehensive information on potential NK cell receptors for targeting NK cell-based immunotherapy.

Sr No.	Type of Receptors	Potential Targets	Ligands	Function	Ref.
1.	Activating receptors	NKG2D	Retinoic acid early induced transcript-1 (RAE-1) H60 UL-16 binding proteins like transcript-1 MHC-I chain-related protein A ULBPs	Activation, proliferation, and expansion of the immune cells	[71]
2.		DNAM-1	Nectin-like proteins PVR proteins	Enhances the cytotoxic activity	[72]
3.		NCR	NKp44L HSPG Heparin Vimentin B7-H6 BAG-6 PCNA	Could stimulate and inhibit NK cell activity based on the ligand	[73]
4.		CD16	Fc IgG	Promotes cytokine production and enhances cytotoxicity	[74]
5.		NCAM/ CD56	-	Enhances cytokine production, IFN- γ production and enhances NK cell proliferation	[75]
6.		CD96	PVR proteins	Enhances recognition of tumor cells, increases expression of cytotoxic granules	[76]
7.		CXCR1	CXCL8 (IL-8)	Enhances homing of immune cells	[77]
8.		CX3CR1	CX3CL1 Fractalkine	Enhances homing of immune cells	[78]
9.		CD11	-	Maturation biomarker for NK cells	[79]
10.		CCR7	CCL19 CCL21	Facilitates homing of NK cells to the lymph nodes	[80]
11.	Inhibitory receptors	CD94/ NKG2A	HLA-E HLA-A	Inhibits cytotoxic ability	[81]
12.		TIM-3	Galectin-9 CEACAM-1 PtdSer HMGB1	NK cell dysfunction and induces NK cell exhaustion	[82]
13.		TIGIT	Nectin-like proteins PVR proteins	Inhibits cytotoxic ability	[83]
14.		KIRs	HLA-A HLA-B HLA-C	Inhibits cytokine release	[84]
15.		LAG-3	MHC-II	Inhibits immune cell activation and cytokine production	[85]
16.		PD-1	PD-L1 PD-L2	Functional defects in NK cells, T cells	[86]

**Table 2 cancers-14-05438-t002:** Comprehensive information on nanoparticle-based immunomodulation for enhancing NK cell activity.

Sr No.	Nanoparticles	Target NK Cell	Ligand	Mechanism of Activation	Effect on Tumor Cells	Ref.
1.	RGD peptide-tagged polyethylene glycol loaded with doxorubicin and diselenium nanoparticle	-	-	Activation of NK cells via seleninic acid	Increased antitumor response Facilitated release of doxorubicin	[103]
2.	*Lactobacillus brevis* enriched selenium nanoparticle			*L. brevis* induced NK cell activation and assisted with local cytokine production Elevated levels of IL-17 and IFN-γ levels	Reduced tumor metastasis observed in liver cells	[104]
3.	Pemetrexed and cytosine-containing diselenide-loaded nanoparticles	-	-	Activation of NK cells by suppressing HLA-E	Reduction in the tumor volumes in MDA-MB-231 tumor-bearing mice	[108]
4.	Folate-containing liposomes, manganese protoporphyrin	Folate receptor	Folate	Facilitated antitumor function of macrophage M1 to activate NK cells and dendritic cells in the tumor microenvironment	Folate-MnP nanoparticles produced excellent sonodynamic therapy in triple-negative breast cancer models	[109]
5.	Calcium carbonate-coated gold nanostars with chlorine e6 photosensitizer	-	-	Increased immune response and antitumor cytokines like TNF-α, IFN-γ, Granzyme A, perforin, FASL, TRAIL, and IL-18	Combined immunotherapy and photodynamic therapy decreased the tumor volume in A549 tumor-bearing mice	[110]
6.	Lipid nanoparticles conjugated to CpG oligonucleotides, mannose, and H22 hepatoma lysate	-	-	Increased levels of IFN- γ The proliferation of NK cells and cytotoxic T cells	Showed excellent tumor growth inhibition and improved survival time in mice with H22 hepatocellular carcinoma	[111]
7.	PLGA nanoparticles encapsulated TLR7/8 agonist	TLR 7/8 receptor	TLR7/8 agonist	CD70 enhanced NK cell activation Enhanced pro-inflammatory cytokines	Improved survival was observed in B16F10-OVA tumor-bearing mice	[112]
8.	TGF-β and IL-2 loaded nanolipogels	TGF-β receptor	TGF-β	Expansion and proliferation of NK cells	-	[101]
9.	Camptothecin-loaded cyclodextrin-based polymers	-	-	Activation of NK cells and T cells	Reduced tumor growth in F1B-tag transgenic mice	[113]
10.	Magnetic nanoparticles	-	-	Increased expression of CCR4 and CXCR4	Reduced the growth of tumors in MDA-MB-231 triple-negative breast cancer-bearing models	[114]
11.	Graphene oxide nanoclusters loaded with anti-CD16	CD16	Anti-CD16 antibody	Increased the levels of IFN-γ and enhanced degranulation in NK cells	-	[115]
12.	CAR-NK cell loaded with paclitaxel	CD16, HER-2 receptor	Anti-HER-1 antibody, Anti-CD16 antibody	Enhanced NK cell infiltration	CAR-NK cells enhanced the tumor-killing efficacy	[116]
13.	PDMAEMA-PTPN6 conjugated atezolizumab	PD-L1	Atezolizumab	Upregulation of NK cells and T cells within the tumor	Prolonged survival in HCT116 tumor-bearing mice	[117]
14.	Magnetic nanoparticle conjugated with a peptide derived from PD-1	-	-	Enhanced the cytotoxic ability of NK cells	Encouraged caspase 3, and caspase 8 for apoptosis	[118]
15.	Immunoliposomes loaded IL-2 and anti-CD137	CD137	Anti-CD137 antibody	Stimulated cytotoxic lymphocytes to infiltrate the solid tumors, cytokine production, and increased granzyme expression	Reduction in tumor cells	[119]
16.	Mesoporous ruthenium nanoparticles conjugated bispecific antibodies (SS-Fc, anti-CD16, and anti-CEA)	CD16 and CEA	Anti-CD16 and anti-CEA	Activated NK cells	Induced cytotoxicity by apoptosis and necrosis in tumor cells	[106]
17.	Trispecific (antibodies of α-CD16, α-4-1BB, and α-EFGR) nanoengagers	CD16, EGFR	α-CD16, α-4-1BB, and α-EFGR	Activated NK cells	Induced cytotoxicity by apoptosis and necrosis in tumor cells	[120]
18.	TRAIL and anti-NK1.1 antibody decorated liposomes	NK1.1 receptor, death receptor	An anti-NK1.1 antibody, TRAIL	Activated NK cells	Increased apoptosis in tumor cells in inguinal lymph nodes and reduced metastases	[121]
19.	cdGMP and MPLA encapsulated nanoparticle	-	-	Encouraged homing, produced interferons	Remodulation of the tumor microenvironment by aiding immune cell infiltration	[122]
20.	Manganese dioxide nanoparticles	-	-	Activated NK cells	MnO_2_ reversed hypoxia in the tumor microenvironment	[123]
21.	Lipid nanoparticle-loaded STING agonist	STING receptor	STING agonist	Increased the expression of CD3, CD4, NK1.1	Restriction of lung metastasis in mice model	[124]
22.	Selenium containing nanoparticle loaded pemetrexed	-	-	Seleninic acid activated immunocompetence in NK cells	Tumor volume reduction in lung cancer model	[125]
23.	Selenopeptide nanomedicine	-	-	Selenopeptide activated NK cells	Improvement in antitumor efficacy	[126]
24.	4,4′,4′′,4′′′-(porphine 5,10,15,20-tetrayl) tetrakis (benzoic acid) (TCPP)-loaded nanoparticle	-	-	Enhanced pro-inflammatory M1 macrophages polarization to produce antitumor immunity	NK cell infiltration could eliminate the primary tumor and produce an abscopal effect on distant tumor sites	[127]

**Table 3 cancers-14-05438-t003:** Comprehensive information on nanoparticle assistance to enhance homing of NK cells.

Sr No.	Nanoparticles	Target NK Cell	Ligand	Mechanism of Activation	Effect on Tumor Cells	Ref.
1.	Iron oxide nanoparticles on the surface of primary NK cells	-	-	Increased the expression of granzymes and perforins Increased homing of NK cells in the presence of a magnetic field in the 3D microfluidic device	Percentage means killing was 33.2% with NK:IONP without a magnet, while it was increased to 58.3% in the presence of a magnet	[89]
2.	Iron oxide nanoparticles labelled on NK cells	-	-	Increased homing in the presence of a magnetic field	Nanoparticles showed high cytotoxicity against A549 cells	[137]
3.	Magnetic PLGA microspheres containing recombinant IFN-γ and iron oxide nanocubes			Increased infiltration of NK cells and other immune cells Activation of NK cells via IFN-γ signals	Microspheres assisted in vivo MRI imaging to locate hepatic tumor pre- and post-treatment	[136]
4.	Dextran-coated serine protease granzyme-B functionalized superparamagnetic iron oxide nanoparticles	mHsp70	Granzyme B	Increased expression of granzyme B	Perforin-independent apoptosis in GL261 orthotopic tumor model	[138]
5.	IL-12 bound gold nanoparticle tagged with homing peptide	αvβ3-integrin receptors	isoAsp-Gly-Arg homing peptide	Activation of NK cells via IL-12 Infiltration of immune cells	Reduction in tumor in WEHI-164 fibrosarcoma and the TS/A adenocarcinoma model	[139]
6.	IL-12-loaded chitosan nanoparticles	-	-	Infiltration of immune cells Activation of NK cells via IL-12	Reduced tumor volume in CT26 liver metastasis mice model	[140]
7.	Anti-GD2 antibody tagged gold nanoparticle	GD-2 receptor	Anti-GD2 antibody	Increased NK cell-mediated antitumor activity	Antibody-dependent cell-mediated cytotoxicity (ADCC) in NB1691 and M21 cells	[141]

**Table 4 cancers-14-05438-t004:** Comprehensive information on nanoparticle-based post-translational gene silencing for activation of NK cells.

Sr No.	Nanoparticles	Target NK Cell	Ligand	Mechanism of Activation	Effect on Tumor Cells	Ref.
1.	Manganese dioxide nanoparticle containing siRNA-TGFBR2	TGFBR2 silencing	siRNA	Silencing of TGFBR2 activated NK cells Increase in the expression of IFN-γ	Changed immunoresponsiveness of tumor microenvironment	[143]
2.	EpCAM targeted cationic liposomes containing si-CD47 and si-PD-L1	EpCAM, CD47, PD-L1	si-CD47, si-PD-L1	Increased the percentage of NK cells Increased the expression of IFN-γ and IL-6	Reduction in tumor growth and lung metastasis in 4T1 tumor-bearing mice	[144]
3.	Cationic lipid-assisted nanoparticles encapsulated with siCD155	CD155	si-CD155	Activation of NK cells and T cells	Reduced cell proliferation in melanoma and reprogrammed the tumor microenvironment	[145]
4.	miRNA-loaded NK cell-derived exosomes	-	let-7a	Increased expression of CXCR4	let-7a enhanced apoptosis in human neuroblastoma cells	[95]
5.	miRNA-186 loaded NK cell-derived exosomes	-	miRNA-186	Promoted NK cytotoxicity and blocked immune escape	Increased apoptosis in *MYCN* gene-amplified neuroblastoma cells	[146]

**Table 6 cancers-14-05438-t006:** Comprehensive information on nanoparticles activating NKG2D receptor.

Sr No.	Nanoparticles	Target NK Cell	Ligand	Mechanism of Activation	Effect on Tumor Cells	Ref.
1.	Super magnetic nanoparticle Zn-CoFe_2_O_4_@Zn-MnFe_2_O_4_	NKG2D, VEGFR	UL16-binding protein	Activated NK cells released TNF-α and IFN-γ	Decreased liver cancer cell viability	[154]
2.	Dendritic cell-derived exosomes	NKG2D	NKG2D ligands	Increased expression of IL-15Rα, CD69, IFN-γ	Decreased tumor volume in human advanced melanoma patients	[155]
3.	Glioblastoma cells pre-exposed NK cell-derived exosomes	NKG2D	-	Activation of NKG2D via IL-2 and IL-5 Activated CD 56 and KIR2DL2 receptor	Drastic reduction in the tumor volume after 4 weeks of treatment in glioblastoma mice model	[156]
4.	TGF-β inhibitor and selenocysteine-containing nanoemulsion	NKG2D	-	Activation of NKG2D via restricting TGF-β/TGF-β RI/Smad2/3 signaling pathway Selenocysteine suppressed PD-1 expression	Increased MDA-MB-231 triple-negative breast cancer cell lysis by 13.8-fold	[153]

## Data Availability

Not applicable.

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
