# Peer review of "Nanoparticle Enhancement of Natural Killer (NK) Cell-Based Immunotherapy"

_cancers, 2022, doi:10.3390/cancers14215438_

Round 1
Reviewer 1 Report
Excellent ms. Acceptable in the current form
Author Response
Response to Reviewer 1
Reviewer 1
Excellent ms. Acceptable in the current form
Answer: We thank the Reviewer for appreciating our work and for her/his encouraging comments on our review article.

Reviewer 2 Report
The authors of the manuscript entitled: ‘Nanoparticle Enhancement of Natural Killer (NK) Cell-based Immunotherapy.’ focused their review on the literature of NK cells, trying the point out the role of NK cells, their mechanism of action in killing tumor cells, and the receptors which could serve as potential targets for signaling. Moreover, they also tried to highlight the role of nanoparticles in NK cell activation and cytotoxicity against cancer.
Despite the fact that the authors tried to include very recent information regarding this topic, there are some important issues that needed to be addressed.
Major Comments:
1. The Graphical abstract and figure 3 are almost identical, therefore it is recommended to change the Graphical abstract, since figure 3 is already adapted from others.
2. Since this topic is of high importance, it is recommended to add a table on current clinical trials using that type of immunotherapeutic approach.
Minor Comments:
1. An extensive spell check is highly recommended.
2. Please change the INF-gamma into IFN-gamma through out the manuscript.
3. Please state in text ‘ICI’ as immune checkpoint inhibitors before using the abbreviation.
4. The paragraph (line 175) describing the CAR-T is confusing to the reader. Please re-phrase. It will raise many questions since many CAR-T therapies are showing very promising results especially in blood cancers.
5. Line 196: please elaborate on the first part: what are the strategies used to activate the NKs. Epigrammatically state a few. (i.e., delivery of cytokines)
6. Table 2 should be reported in the text (like rest of the Tables).
7. Last but not least, the references (150-151) do not describe RNAi for gene depletion, but addition of microRNAs, thus do not fit in Table 4 (change title of the Table or/and explain in text).
Author Response
Response to Reviewer 2
Reviewer 2
We thank the learned reviewer for providing suggestions that were very useful in the improvement of the quality of the manuscript. We are happy to attend to all his/her queries and revise the ms suitably by incorporating his/her suggestions. Answers to all his/her queries and modifications incorporated are listed below (For your information, the changes made to the document are highlighted in yellow in the manuscript).
Major Comments:
- The Graphical abstract and figure 3 are almost identical, therefore it is recommended to change the Graphical abstract, since figure 3 is already adapted from others.
Answer: We thank the reviewer for this suggestion, and we have modified the graphical abstract. (Page no-2, line no-38-39)
- Since this topic is of high importance, it is recommended to add a table on current clinical trials using that type of immunotherapeutic approach.
Answer: As per the reviewer’s suggestion, nanoparticle-based NK cell therapy has only one study that has entered clinical trial, which has been mentioned in the manuscript (Page no-7, line no-207-217).
Minor Comments:
- An extensive spell check is highly recommended.
Answer: As per the suggestion from the reviewer, we have checked the entire manuscript and corrected the errors.
- Please change the INF-gamma into IFN-gamma throughout the manuscript.
Answer: As suggested by the reviewer, we have modified the abbreviation throughout the manuscript (Page no-3, line no-99), (Page no- 6, table no- 5th point), (Page no- 8, line no-236), (Pages no-8-9, table no- 2, 2nd,5th, 6th, and 11th point), (Page no- 10, line no- 273), (Page no- 11, table no-3, 3rd point), (Page no-11, line-293), (Page no-12, line no-302), (Page no-12, table no- 4, 1st and 2nd point), (Page no-15, table no-6, 1st, and 2nd point), and (Page no-16, line no-456).
- Please state in text ‘ICI’ as immune checkpoint inhibitors before using the abbreviation.
Answer: As suggested by the reviewer, we have incorporated the abbreviation (Page no-4, lines 132-133).
- The paragraph (line 175) describing the CAR-T is confusing to the reader. Please re-phrase. It will raise many questions since many CAR-T therapies are showing very promising results especially in blood cancers.
Answer: As suggested by the reviewer, we have re-written the sentence (Page no-5, line no-168-169).
- Line 196: please elaborate on the first part: what are the strategies used to activate the NKs. Epigrammatically state a few. (i.e., delivery of cytokines)
Answer: As per the suggestion from the learned reviewer, we have elaborated on the first part of the strategy. (Page no-5, line no-187-188).
- Table 2 should be reported in the text (like rest of the Tables).
Answer: As suggested by the reviewer, we have added table 2 in the intext (Page no-8, line- 231).
- Last but not least, the references (150-151) do not describe RNAi for gene depletion, but addition of microRNAs, thus do not fit in Table 4 (change title of the Table or/and explain in text).
Answer: As suggested by the reviewer, we have revised the title of table 4 to “Comprehensive information on nanoparticle-based post-translational gene silencing for activation of NK cells” (Page no-12, line-328-329).

Reviewer 3 Report
The authors reviewed different areas where nanotechnology enhances immunotherapy in the context of NK cell immunomodulation. I consider this review has complete and updated information in this field, I didn't find many coincidences in this area. There is any current therapy in development by any company? If yes, It will be great to add the status of those trials; but I think many of those therapies in the tables are part of different groups' research (also it's great to know what is going on). The authors many can extend the information about those therapies including potential pitfalls, or limitations.
Author Response
Response to Reviewer 3
Reviewer 3:
The authors reviewed different areas where nanotechnology enhances immunotherapy in the context of NK cell immunomodulation. I consider this review has complete and updated information in this field, I didn't find many coincidences in this area. There is any current therapy in development by any company? If yes, It will be great to add the status of those trials; but I think many of those therapies in the tables are part of different groups' research (also it's great to know what is going on). The authors many can extend the information about those therapies including potential pitfalls, or limitations.
We thank the learned reviewer for appreciating our work and also for providing several suggestions for improving the quality of our manuscript. We are happy to incorporate the suggestions made, and we sincerely believe that the revised manuscript is now suitable for publication. Answers to all his/her queries and modifications incorporated are listed below (For your information, the changes made to the document are highlighted in yellow in the manuscript).
Answer: As per the suggestion from the learned reviewer, we have incorporated the limitation of the nanotechnology-based NK cell therapy section. (In section 3) (Page no-15-16, line no-411-435).

Reviewer 4 Report
The manuscript describes the improved anti-tumor effect of NK cell therapy mediated by nanoparticles. Application of nanoparticles in modulating immune cells is an interesting area and getting more attention in cancer immunotherapy. Overall, this is a well-written review article. In the manuscript, characteristic features of improved nanoparticle-mediated anti-cancer NK cells are well summarized. I would like to add some comments, which, in my opinion need to be addressed to improve the manuscript.
1. In the figure1, the ‘immunoediting’ is schematically well described. The explanation in the introduction section seems to be unnecessarily long. It is to recommend rather to add updates from new published studies for strengthening the manuscript.
2. Graphical abstract is a duplicate of figure3. Figure 3 could be removed.
3. Additional figure or a figure replacing figure4 that shows a general working model of the nanoparticle-assisted NK cell-based immunotherapy (e.g. a scheme explaining the CAR-T therapy: remove blood from patient to get T cells -> make CAR T cells in the lab -> grow millions of CAR T cells -> Infuse CAR T cells into patient -> CAR T cells bind to cancer cells and kill them) would be a great help to better understand the therapy.
4. It is to recommend to add a short section (maybe in 2.6) and describe about the possible limitation of the NK cell-based therapy assisted by nanoparticles, different efficacies in various cancer types.
Author Response
Response to Reviewer 4
Reviewer 4:
The manuscript describes the improved anti-tumor effect of NK cell therapy mediated by nanoparticles. Application of nanoparticles in modulating immune cells is an interesting area and getting more attention in cancer immunotherapy. Overall, this is a well-written review article. In the manuscript, characteristic features of improved nanoparticle-mediated anti-cancer NK cells are well summarized. I would like to add some comments, which, in my opinion need to be addressed to improve the manuscript.
We thank the Reviewer for her/his encouraging comments and recommendation for publication in Cancers. Answers to all his/her queries and modifications incorporated are listed below (For your information, the changes made to the document are highlighted in yellow in the manuscript).
- In the figure1, the ‘immunoediting’ is schematically well described. The explanation in the introduction section seems to be unnecessarily long. It is to recommend rather to add updates from new published studies for strengthening the manuscript.
Answer: As per the learned reviewer’s suggestion, we have reduced the immunoediting content in the introduction section and also included some new references from recent articles. (Ref no- 26 and 32) (Page no-3, lines 91, 99).
- Graphical abstract is a duplicate of figure 3. Figure 3 could be removed.
Answer: As per the suggestion from reviewer 2, we have modified the graphical abstract, and thus, there will not be a repetition of figure 3 in the manuscript.
- Additional figure or a figure replacing figure4 that shows a general working model of the nanoparticle-assisted NK cell-based immunotherapy (e.g. a scheme explaining the CAR-T therapy: remove blood from patient to get T cells -> make CAR T cells in the lab -> grow millions of CAR T cells -> Infuse CAR T cells into patient -> CAR T cells bind to cancer cells and kill them) would be a great help to better understand the therapy.
Answer: As per the suggestion from the reviewer, we have replaced the schematic figure representing CAR-NK cell therapy instead of figure 4 (previous figure 4). (Page 13, line no-337-341).
- It is to recommend to add a short section (maybe in 2.6) and describe about the possible limitation of the NK cell-based therapy assisted by nanoparticles, different efficacies in various cancer types.
Answer: As per the suggestion from the learned reviewer, we have incorporated the limitation of the nanotechnology-based NK cell therapy section. (In section 3) (Page no-15-16, line no-411-435).
